# Veterinary Big Data: When Data Goes to the Dogs

**DOI:** 10.3390/ani11071872

**Published:** 2021-06-23

**Authors:** Ashley N. Paynter, Matthew D. Dunbar, Kate E. Creevy, Audrey Ruple

**Affiliations:** 1Department of Biology, College of Arts and Sciences, University of Washington, Seattle, WA 98195, USA; apaynt@uw.edu; 2Center for Studies in Demography and Ecology, University of Washington, Seattle, WA 98195, USA; mddunbar@uw.edu; 3Department of Small Animal Clinical Sciences, College of Veterinary Medicine, Texas A&M University, College Station, TX 77843, USA; kcreevy@cvm.tamu.edu; 4Department of Public Health, College of Health and Human Sciences, Purdue University, West Lafayette, IN 47907, USA

**Keywords:** big data, personalized healthcare, companion animal medicine, comparative medicine, one health

## Abstract

**Simple Summary:**

Big data has created many opportunities to improve both preventive medicine and medical treatments. In the field of veterinary medical big data, information collected from companion animals, primarily dogs, can be used to inform healthcare decisions in both dogs and other species. Currently, veterinary medical datasets are an underused resource for translational research, but recent advances in data collection in this population have helped to make these data more accessible for use in translational studies. The largest open access dataset in the United States is part of the Dog Aging Project and includes detailed information about individual dog participant’s physical and chemical environments, diet, exercise, behavior, and comprehensive health history. These data are collected longitudinally and at regular intervals over the course of the dog’s lifespan. Large-scale datasets such as this can be used to inform our understanding of health, disease, and how to increase healthy lifespan.

**Abstract:**

Dogs provide an ideal model for study as they have the most phenotypic diversity and known naturally occurring diseases of all non-human land mammals. Thus, data related to dog health present many opportunities to discover insights into health and disease outcomes. Here, we describe several sources of veterinary medical big data that can be used in research. These sources include medical records from primary medical care centers or referral hospitals, medical claims data from animal insurance companies, and datasets constructed specifically for research purposes. No data source provides information that is without limitations, but large-scale, prospective, longitudinally collected data from dog populations are ideal for further research as they offer many advantages over other data sources.

## 1. Introduction

The era of big data has opened up many new opportunities in preventive care, chronic disease management, and treatment optimization. It has also allowed for a new model of medicine to be employed, that of personalized or precision medicine. In this model of human healthcare, decisions about medical interventions and other treatments are tailored to an individual patient based upon their predicted risk of developing disease or their predicted response to therapy [1]. This has been enabled primarily through the adoption of electronic health record systems by healthcare providers, which allow for the construction of detailed longitudinal data about large populations of patients over long periods of time. These data frameworks can be interrogated to discover the combinations of risk factors that lead to disease outcomes and deliver personalized disease risk profiles for individual patients [2].

Data-related problems in human healthcare such as data integration, wrangling, ease of use, and interpretability are similar to those encountered in other domains. However, there are a number of domain-specific challenges encountered in human healthcare disciplines that are unique to these data frameworks. One example of this is the use of unstructured data, such as patient notes and interpretations of diagnostic tests, which contain rich information that can provide valuable insights at both the individual patient and population levels, but the heterogeneity, variability, and diversity of these data make them difficult to access if analyzed in a controlled manner [3]. Another challenge lies in the issues of privacy and security in human healthcare, which have drawn significant attention in recent years, but are especially important in healthcare settings because of concerns related to the introduction of the HIPAA Privacy Act, which declared medical information, including electronic medical records, to be protected health information covered under the Privacy Rule [4,5]. In fact, the sheer magnitude of the size of medical big data has increased to such a level that new storage technology systems have been developed to capture, manage, and process big data. However, issues with privacy control, technical vulnerabilities, security for authorization and verification, data management, and confidentiality still exist [6].

Many of these challenges surrounding medical data and use are avoided in the field of veterinary medical big data, as veterinary patient records are not considered protected health information and are thus not included in the Privacy Rule. However, naturally occurring diseases in companion animals are often similar, if not identical, to human diseases in terms of disease etiology, progression, and treatment response [7]. Dogs, in particular, provide an ideal model for translational medicine as they have the most phenotypic diversity and known naturally occurring diseases of all land mammals other than humans [8]. Among those diseases, we have identified more than 400 inherited disorders in dogs that are relevant to humans, which is likely due to the fact that we share ~650 Mb of ancestral genetic sequence with our dog companions [9]. In addition, dogs share both our physical and chemical environments and live in more than 63 million households in the United States [10]. Disease outcomes as varied as cancer (e.g., lymphoma, osteosarcoma), osteoarthritis, spinal cord disorders (e.g., thoracolumbar intervertebral disk herniation, spina bifida), eye disorders (e.g., keratoconjunctivitis sicca), cardiomyopathies (e.g., dilated and hypertrophic cardiomyopathies), and infections—including those caused by antimicrobial-resistant organisms—are all shared by our dog companions [7,11]. Furthermore, the healthcare system for dogs in the United States is sophisticated and over USD 40 billion is spent annually on dog health care, second only to the level of healthcare received by humans [12,13].

Thus, veterinary medical big data may very well represent an underutilized resource in medical research. Using data derived from dogs whose owners are seeking medical care for naturally occurring diseases may also have less ethical concerns than research involving animals in which the disease of interest is induced [7]. However, despite the reduced concerns related to privacy and security, accessing veterinary medical big data is not without challenges. In this review, we will describe the veterinary medical data sources available for research as well as the opportunities and challenges related to each.

## 2. Data Sources for Companion Animal Health Information

There are several sources of veterinary medical big data related to companion animal health information that have been utilized for research purposes. The majority of these data sources are comprised of medical records from either primary medical care centers or referral hospitals. Another data source for animal health information comes from animal insurance companies, which typically track medical claims data on privately owned animals. In the United States, approximately 2.5 million pets were insured in 2019, which was a growth of 16.7% over the total number of pets insured in the prior year, but still only represented about 2.5% of the total pet population in the US [14]. Insurance coverage rates for companion animals in the United Kingdom are approximately 25% of the total population and it is estimated that 90% of the dog population in Sweden are covered by an insurance policy [15]. Other datasets have been constructed specifically for research purposes and therefore, often only include animals that meet explicit inclusion criteria, but might still be useful for translational research.

### 2.1. Medical Record Datasets

One of the largest veterinary practices in the world is Banfield Pet Hospital, a privately owned company that consists of more than 1000 individual clinics. As of 2019, Banfield Pet Hospital had collected clinical data from more than 2.5 million dogs from 43 states in the US [16]. All Banfield hospitals use a single proprietary medical record software system to upload and centrally store electronic information related to the companion animals seen in individual hospital locations. These records include information such as laboratory test results, physical examination findings, diagnoses, treatments, procedures performed, and demographic information about the dog patients and their owners [17]. In addition, these data include geocoded information about the home locations of the animals seen at Banfield hospitals. This has allowed researchers to pair electronic record data with existing ecological data to monitor infectious disease spread in animals in the US [18].

One drawback of utilizing Banfield medical data for translational medicine analyses is that these data do not consistently contain behavioral information such as type and quantity of outdoor activities undertaken, exercise history, diet composition, and temperament of the pet [19]. Diagnoses contained in the dataset may include both those suspected by the veterinarian and those with diagnostic test confirmation [17]. However, the large number of pets with diagnoses and veterinarians employed at these clinics minimizes the risk of systematic misclassification errors occurring. Perhaps most importantly, these data are owned by Banfield Pet Hospital and access to these data appears to be limited.

Another source of veterinary health information is the Veterinary Medical Database (VMDB), the oldest companion animal health database in the United States, which provides veterinary medical datasets to researchers at little or no cost. This database was started in 1964 by the National Cancer Institute and includes patient data contributed from 26 university teaching hospitals in the United States and Canada, and contains over 7 million records from all species [20]. The VMDB diagnoses data were originally coded using the Standard Nomenclature of Veterinary Diseases and Operations (SNVDO), but have been coded in Systematized Nomenclature of Medicine—Clinical Terminology (SNOMED CT) since 1996 [20]. It is important to note that this hierarchical coding structure was originally created for diagnoses in human patients rather than veterinary patients, but use of these coding systems allows for searches to be conducted in either broad disease categories or using only specific diagnoses [21].

A limitation of this dataset is the narrow availability of information related to each medical record as only a coded abstract from each patient hospital visit is included in the dataset. Data contained are limited to: the institution where the animal was seen, the species and breed of the animal, signalment (age, weight, sex), hospital discharge date, patient number (as assigned by the hospital), if the hospital visit was the first or a recheck appointment for a previously diagnosed condition, diagnoses, and the postal code of the client [20]. Additionally, disease prevalence estimates derived from this dataset are likely influenced by referral bias as veterinary teaching hospitals provide highly specialized and advanced animal care and so the underlying population from which the VMDB data are derived is biased towards sick animals and more serious and rare diseases are overrepresented in this database [21].

The Small Animal Veterinary Surveillance Network (SAVSNET) is a database of electronic health and environmental data on companion dogs in the United Kingdom. SAVSNET was initially formed in 2008 by the British Small Animal Veterinary Association (BSAVA) and the University of Liverpool [22]. Currently, the database is entirely run by the University and as of 2016, it has been expanded in the hopes of being used for translational analyses. A major benefit of SAVSNET is its dedication to enhancing understanding of the impact of antimicrobial resistance, climate, environmental risk factors, and infections on overall health.

SAVSNET is comprised of two data sources, SAVSNET-Lab and SAVSNET-Vet. The former surveils records from veterinary diagnostic laboratories in order to monitor disease statistics of canines across the UK. The latter is comprised of information on symptomology from veterinary surgeons recorded at practices at the end of each appointment. SAVSNET releases web reports that detail current disease statistics in the small animal population in England and Wales. SAVSNET data are accessible through an application process and, if approved, researchers are paired with a “data chaperone” who is highly knowledgeable about the database and collection methodology and can recommend approaches for statistical analyses and data handling [23].

The Veterinary Companion Animal Surveillance System (VetCompass), which began collecting clinical data from primary practices in the United Kingdom in 2009, now holds data on more than 15 million animals collected from over 1800 veterinary practices across the UK, nearly a third of all UK veterinary practices [24]. VetCompass began collecting clinical data in Australia in 2016 and pilot projects are being completed in Spain, Germany, and New Zealand [25,26]. These data are actively collected and considered to be more representative of the general pet population than data collected only from referral hospitals; geographic and temporal information are included, which allows for analyses of disease trends over both space and time [25,26].

Clinical data from each participating practice are uploaded automatically to the VetCompass database and require no additional work on the part of the hospital staff or veterinarians [26]. Extraction of these data has not yet been fully automated though, which can result in data access limitations. Another limitation is that these data do not include behavioral, environmental, or dietary information about these patients, all of which can be important factors in the etiology of disease. Open access VetCompass data are available through request, though typically only for use in academic settings [24].

### 2.2. Insurance Datasets

Data collected by veterinary medical insurance providers can be used as a secondary data source for translational analyses. One of the biggest advantages of insurance datasets is that they are large and high statistical power can be achieved for many types of analyses. These types of datasets include varying amounts of information about the total population of insured pets included in the dataset, but uniformly include detailed address information about individual owners. This allows for more robust interpretations about the impact of geographic factors on health outcomes to be made than might be possible using only medical records [27]. However, the coding system used by individual insurance companies may allow for only one diagnostic code to be entered for each veterinary visit, which can result in a reduction in data completeness [27]. In addition, these datasets may not distinguish between death from natural causes and death from euthanasia, which should be considered when constructing time to event analyses [28].

Two well-established insurance databases, Pet Protect from the UK and Agria from Sweden, have been utilized for epidemiological research on the causes of morbidity and mortality in insured dog populations [28,29]. The dogs in the UK insurance dataset were thought to differ from the total population of dogs in the UK in that the insurance population had overrepresentation of younger and purebred dogs [29]. There was also a lack of histologic confirmation of some disease diagnoses noted by the investigators [29].

The Swedish insurance dataset has been validated and shown to have adequate agreement with medical records obtained from veterinary practices that provided medical care to the animals included in the dataset [30]. The large proportion of dogs covered by insurance policies in Sweden is a benefit in that the insurance datasets are likely more representative of the total dog population in the country than the datasets available in the US and the UK, thereby increasing the external validity of research conducted using this population of dogs. The Swedish dataset held by the insurance company Agria has been utilized by several research teams over many decades, too, so there is a long record of open access to these data which may now provide insights into changes in disease trends over time that would not be identifiable in other datasets.

### 2.3. Research Datasets

Other datasets have been constructed to investigate health and disease trends in particular populations of dogs. One such dataset is part of the Golden Retriever Lifetime Study (GRLS), a prospective study restricted to a single cohort of approximately 3000 Golden retriever dogs located throughout the US. Despite the limitations created by the small number of individual animals included in this dataset, there are some significant benefits to using these data for translational studies. For instance, these data were collected longitudinally and at regular intervals and the dataset contains information on not only the health of animals collected directly from their veterinarian and including laboratory values, but also extensive environmental and behavioral health data collected from their owners [31]. Despite the burden placed on owners to track and report data about their dogs, this work has had a high level of study compliance and has resulted in a robust dataset that includes information about individual dogs from the time of enrollment at less than 2 years of age until their death [32].

An added advantage to GRLS is that biological samples have been collected from these dogs on at least an annual basis [31]. These samples include whole blood, serum, urine, hair and toenail clippings, feces, and tissue samples, all of which were stored in a biobank at the time of collection and are available for interrogation by investigators [31]. These samples offer a unique opportunity to investigate environmental exposures over the lifetime of the dogs and their relationships to multiple disease outcomes, which could lead to insights that will benefit both animal and human health.

Another research platform, Dogslife, is collecting data directly from owners about the health and lifestyle of their UK Kennel Club-registered Labrador retrievers [33]. The study population is restricted to purebred Labrador retriever dogs born on or after 1 January 2010 and all data collection is performed using a website interface [33]. Owners of enrolled dogs are requested to complete a questionnaire about their dog’s health and welfare monthly for the first year of the dog’s life and at 3-month intervals thereafter. The compliance with questionnaire requirements is reasonably high, with nearly 80% of the data entries being complete at the time of the first descriptive evaluation of the cohort [33]. By 31 December 2013, there were more than 4000 dogs included in this dataset and enrollment is ongoing [34]. However, analysis of questionnaire data has shown that many owners make mistakes when reporting. This includes typos, using the wrong unit of measurement and inaccurately measuring and weighing the dog. The DogsLife team has created “data cleaning” protocols as part of data analysis to account for some of the bias introduced by human error [35].

## 3. The Dog Aging Project

A long-term longitudinal study of companion dogs, the Dog Aging Project (DAP), was initiated in 2019 with the express purpose of collecting data that could be used in translational medical research. Dogs of any breed, size, and age are eligible for participation in DAP, though enrollment is currently limited to the 50 US states. This project is an Open Science study and investigators unaffiliated with the project are able to access these data as well as propose ancillary studies that can build upon the foundation created by the DAP team. It is anticipated that the total number of dogs enrolled in this project will exceed 100,000 as enrollment is ongoing and there is no limit to the total number of participants that will be included in the project. Thus, this dataset represents the largest primary data source of health information in dogs collected explicitly for use in translational medical research.

### 3.1. Health and Life Experience Survey

Comprehensive information about the dog’s health and environment is collected at the time of enrollment in the study through the Health and Life Experience Survey (HLES). This survey is administered through an online portal and is separated into ten sections. Data collected through this instrument include detailed information about the dog’s breed, age, sex, neuter status, behavior, diet, use of medications, types and amount of physical activity, the indoor and outdoor environments the dog is regularly exposed to, and the dog’s comprehensive health history. The initial HLES data are considered the baseline dataset for each participant in the project. HLES data are updated annually in order to identify changes related to either the dog or their environment as well as collect information about any new medical diagnoses made within the previous year.

### 3.2. Other Types of Data Collected

In addition to the HLES data, all participants are asked to upload copies of the dog participant’s electronic medical record. These records can be used to validate information provided by human participants about their dog’s overall health, specific diagnoses reported, and demographic information. In addition, laboratory values and other items of research interest may be extracted from the medical records and used to help inform specific research objectives.

Approximately 10,000 dogs included in DAP will have whole genome DNA sequencing (WGS) performed. This will add tremendous value to this dataset as genome wide association studies (GWAS) performed in dog populations have been shown to be able to identify functional variants associated with specific morphologic traits as well as disease risk and dog behavior [36]. WGS has also been utilized to identify rare hereditary diseases in dogs as well as isolate variants located in orthologs of human cancer susceptibility genes [37,38,39].

Approximately 1500 of the dogs that have whole genome sequencing data collected will also have information related to their metabolome, methylome, microbiome, and rest/activity cycles collected and included in the dataset. Five hundred of these dogs will also have physical examinations and diagnostic testing performed by veterinary cardiologists and results from echocardiogram, electrocardiogram, and blood pressure measurement tests will be included in the DAP dataset.

These additional data inputs, sequencing, and physiologic measurements will enrich the data available for translational research using these subsets of DAP dogs. Ancillary studies conducted using other DAP dogs may further supplement the data available in the dataset in the future. These ancillary studies will likely be focused on specific research questions and the additional data available will be dependent upon the needs of the investigators.

## 4. The Future of Companion Animal Health Data Collection

The increasing frequency with which companion dogs are being used to study disease outcomes highlights the importance of medical data collected within this population. Each of the datasets described herein (Table 1) provides valuable resources for research. However, use of large-scale epidemiological studies, in which data are collected longitudinally, is ideal for this work as these data allow for temporal analyses to be conducted and for incidence and prevalence of diseases to be estimated.

Secondary datasets, such as those belonging to insurance companies, are useful for this work, but primary datasets do offer some advantages. Because these datasets are constructed with the explicit intent to utilize data for research purposes, there are often both subjective and objective measurements included in the same dataset.

This helps to give a more complete understanding of the dog’s health than data related only to illness (e.g., insurance claims data). Additionally, the increased amount of data collected related to the dog’s environment and potential environmental risk exposures is increased in primary datasets.

Collection of health data across multiple breeds of dogs is ideal in that the increased genetic heterogeneity in a more outbred population of dogs will allow for identification of genes related to a greater number of disease outcomes. In datasets where genomic sequencing is not included, individual dog breeds can be utilized as a proxy for specific genetic information. In these cases, outcomes with varying prevalence in different breeds of dogs can be identified as having a potential genetic or heritable component to the disease etiology.

The initiation of dog health studies, such as DAP, mark a turning point in the utilization of dogs as a model for human health and disease. These large-scale datasets constructed with the express intent to understand the biological and environmental determinants of health and disease in companion dog populations, in order to apply those findings to other species, will become increasingly valuable to medical research as the datasets grow in terms of content and population.

## 5. Conclusions

Use of medical data collected in dog populations will continue to be a rich source of information that can be used to inform our understanding health, longevity, and disease outcomes.

## Figures and Tables

**Table 1 animals-11-01872-t001:** Defining characteristics of datasets available for health research using dog populations.

Dataset Name	Dataset Initiated (Year)	Information about Dataset and Population	Examples of Health Outcomes Investigated Using These Data	Accessibility
Banfield Pet Hospital	1955	Includes medical records from animals seen at over 1000 general practice clinics located in 42 US states, Washington D.C., Puerto Rico, and Mexico	Obesity [40]; Zoonotic infections [18]	Data are accessible through permission gained through direct request
The Veterinary Medical Databases (VMDB)	1964	Over 7 million coded medical records from patients admitted to 1 of 27 university teaching hospitals in the United States and Canada	Aging [41]; Cancer [42]	Data are accessible upon request for a fee; fee is waived for researchers at participating institutions
Small Animal Veterinary Surveillance Network (SAVSNET)	2008	Includes medical records from over 500 veterinary clinics in the UK, laboratory data from 12 veterinary clinical laboratories, climate, and environment data	Zoonotic infections [43,44]	Data access must be approved through a two-step application process; there is a data access fee
The Veterinary Companion Animal Surveillance System (VetCompass)	2009	Includes electronic medical records from more than 15 million animals seen at over 1800 veterinary clinics in the UK	Aging [41]; Cancer [45]; Zoonotic infections [46]	Data used for published research are available through open access, unpublished data are accessible through direct request
Pet Protect Pet Insurance Company (UK)	1985	Insurance claims data from insured dogs in the UK	Cancer [29]	Data are accessible through permission gained through direct request
Agria Pet Insurance Company (Sweden)	1995	Insurance claims data from insured dogs in Sweden.	Diabetes [47]; Kidney disease [48]	Data are accessible through permission gained through direct request
Golden Retriever Lifetime Study (GRLS)	2012	Medical records and questionnaire data for a cohort of more than 3000 Golden retriever dogs in the US	Fertility [49], Obesity [50]	Data are accessible through a formal proposal process
Dogslife	2010	Questionnaire data from owners of over 8000 Labrador retriever dogs in the UK	Gastrointestinal illness [51]	Data are accessible to researchers at participating institutions
Dog Aging Project (DAP)	2019	Medical records and questionnaire data for a cohort of more than 30,000 dogs in the US; genomic, metabolomic, and microbiome data available for some participants	In progress	Curated datasets are available on an annual basis

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
