# Peer review of "Veterinary Big Data: When Data Goes to the Dogs"

_animals, 2021, doi:10.3390/ani11071872_

Round 1
Reviewer 1 Report
The authors have summarized ten different veterinary medical data sources available for research, and the opportunities and challenges related to each of them. This paper could be a very valuable resource for researchers that are not familiar with veterinary data sources, but that would like to start translational research with data from dogs. The value could be clear if the authors include a summary table specifying which data sets are publicly available, which ones are not, and how to access them.
The paper is very well written and has the advantage of summarizing the most useful highlights and limitations of each data set, in an informative and simple way. I was only missing one data set: why are you not including SAVSNET in the list?
One important limitation of the article is that not all the data sets are public and that the authors do not include a persistent unique identifier (PID) for those data sets that are publicly available. It is good that some of them have a reference, but when referring to a data set, it is important to be specific about which version of it you are describing, especially if you are talking about Open Science data sets that are constantly changing.
My recommendation is that the authors add a summary table with the availability of each of the data sets. Please be very specific about how to access them (if possible). Without access references, I find it difficult to evaluate the value of this article.
Author Response
The authors have summarized ten different veterinary medical data sources available for research, and the opportunities and challenges related to each of them. This paper could be a very valuable resource for researchers that are not familiar with veterinary data sources, but that would like to start translational research with data from dogs. The value could be clear if the authors include a summary table specifying which data sets are publicly available, which ones are not, and how to access them.
Response to reviewer: Thank you so much for your review and for your comments. We feel the addition of the table made a great deal of sense and its inclusion has improved the accessibility of the manuscript.
The paper is very well written and has the advantage of summarizing the most useful highlights and limitations of each data set, in an informative and simple way. I was only missing one data set: why are you not including SAVSNET in the list?
Response to reviewer: Thank you for your question. The reason we had chosen to exclude SAVSNET from the manuscript draft was because we were focused on datasets that provided opportunity to use dog data for translational models, which we were interpreting narrowly as naturally occurring models of non-infectious disease. SAVSNET is a robust dataset that is focused more on infectious disease spread and use of animals as sentinels for disease and to track antimicrobial resistance. However, your point is well taken and though SAVSNET did not fit within our more narrow scope we have now added two paragraphs highlighting SAVSNET-lab and SAVSNET-Vet to the updated manuscript.
One important limitation of the article is that not all the data sets are public and that the authors do not include a persistent unique identifier (PID) for those data sets that are publicly available. It is good that some of them have a reference, but when referring to a data set, it is important to be specific about which version of it you are describing, especially if you are talking about Open Science data sets that are constantly changing.
Response to reviewer: Thank you for this comment. The use of PIDs for veterinary medical datasets is not yet adopted or standardized in this way to the authors’ knowledge. Most of the datasets described in this manuscript are ones that are accessed through agreement, rather than through open access, which may be the underlying reason why this has not yet been formally adopted. Even with the Dog Aging Project, which is an open data project, there are yearly curated data releases that are unchanging and thus are referred to by their release number rather than by a PID.
My recommendation is that the authors add a summary table with the availability of each of the data sets. Please be very specific about how to access them (if possible). Without access references, I find it difficult to evaluate the value of this article.
Response to reviewer: This has been done. Thank you so much for the suggestion.
Reviewer 2 Report
This traditional review appears to be conducted by topic experts and it provides an overview of the use of veterinary medical big data in translational medicine that would probably deserve a more rigorous methodological approach, in order to make a complete state of the art in this field and disseminate the best existing examples/project/programs on this very promising reasearch area.
Authors use indifferently the term “medical” and sometimes it is not clear if they actually refer to veterinary or human medicine/care/research fields.
Here few examples where some additional words might be helpful, to improve clarity:
- Lines 39 to 41 -In this model of human healthcare, decisions about medical interventions and other treatments are tailored to an individual patient based upon their predicted risk of developing disease or their predicted response to therapy [1]
- Lines 48 to 50 - Data-related problems in human healthcare such as data integration, wrangling, ease of use, and interpretability are similar to those encountered in other domains. However, there are a number of domain-specific challenges encountered in human healthcare disciplines that are unique to these data frameworks.
- Lines 91 to 92 - There are several sources of veterinary medical big data related to companion animal health information that have been utilized for research purposes.
Overall, apart from the very interesting chapter on the Dog Aging Project, it would be interesting if the Authors could provide additional evidences, showcase other examples/project/programs on the use of dogs’ health big data in human medical research.
With reference to potential additional sources of big data related to companion animal health, it is worth mentioning that other databases already exist, where information on dogs’ health and traceability are stored. Some are managed by government authorities and other from animal welfare NGOs or private companies. The CARO Dog website could represent an additional source of information (https://www.carodog.eu/) to be explored and analysed.
Finally, on the basis of their experience and existing literature in translational medicine, it would be interesting if the Authors could suggest the structure of an “ideal” complete Companion Animal Health Record.
Author Response
This traditional review appears to be conducted by topic experts and it provides an overview of the use of veterinary medical big data in translational medicine that would probably deserve a more rigorous methodological approach, in order to make a complete state of the art in this field and disseminate the best existing examples/project/programs on this very promising research area.
Response to reviewer: Thank you so much for your review and for your comments. We opted not to use a systematic approach to the literature review for this topic as we knew a priori that it was not our intent to describe every dataset ever used in this field as many of them are privately owned and not accessible to all researchers (e.g. – datasets kept by veterinary diagnostic laboratories). The number of datasets we felt did fit the description of being “big data” and at least relatively accessible was small and we felt confident we could identify datasets that had been used for publication in this area without performing a truly systematic search.
Authors use indifferently the term “medical” and sometimes it is not clear if they actually refer to veterinary or human medicine/care/research fields.
Here few examples where some additional words might be helpful, to improve clarity:
- Lines 39 to 41 -In this model of humanhealthcare, decisions about medical interventions and other treatments are tailored to an individual patient based upon their predicted risk of developing disease or their predicted response to therapy [1]
- Lines 48 to 50 - Data-related problems in humanhealthcare such as data integration, wrangling, ease of use, and interpretability are similar to those encountered in other domains. However, there are a number of domain-specific challenges encountered in human healthcare disciplines that are unique to these data frameworks.
- Lines 91 to 92 - There are several sources of veterinarymedical big data related to companion animal health information that have been utilized for research purposes.
Response to reviewer: Thank you for these suggestions, all of which have been made.
Overall, apart from the very interesting chapter on the Dog Aging Project, it would be interesting if the Authors could provide additional evidences, showcase other examples/project/programs on the use of dogs’ health big data in human medical research.
Response to reviewer: The Dog Aging Project is truly the first of its kind in that the dataset is being compiled specifically to inform human medical research. We did add some examples of translational health-related outcomes that have been studied using each of these datasets in Table 1.
With reference to potential additional sources of big data related to companion animal health, it is worth mentioning that other databases already exist, where information on dogs’ health and traceability are stored. Some are managed by government authorities and other from animal welfare NGOs or private companies. The CARO Dog website could represent an additional source of information (https://www.carodog.eu/) to be explored and analysed.
Response to reviewer: The CARO Dog website has a wonderful library of datasets pertaining to companion animals. However, these are typically datasets that are used to track and trace animals and their movements through use of subdermal placement of microchips and do not contain robust health data. Thus, we felt these datasets did not fit with the aims for this manuscript.
Finally, on the basis of their experience and existing literature in translational medicine, it would be interesting if the Authors could suggest the structure of an “ideal” complete Companion Animal Health Record.
Response to reviewer: We do agree that development of ideal complete health records would be a huge gain for this research area. However, there are currently more than 40 commercially available medical record system software packages in addition to the proprietary record systems developed privately by corporately owned clinics or veterinary teaching hospitals. The vast differences between these systems is truly problematic for this field of research and the authors feel that tackling this topic would be deserving of its own publication and would thus be outside the scope of this work.
Round 2
Reviewer 1 Report
This is a useful and interesting article, I have no further suggestions.
Author Response
Thank you so much for taking the time to review this manuscript again!
Reviewer 2 Report
The paper has been improved, Authors might consider to take on board the folllowing additional edits :
- Lines 21 to 22 - Large scale datasets such as this can be used to inform our understanding of human health, disease, and how to increase healthy lifespan.
- Lines 25 to 26 - Thus, data related to dog health presents many opportunities to discover insights into human health and disease outcomes. Here we describe several sources of veterinary medical big data that can be used human in research.
- Lines 27 to 28 - These sources include veterinary medical records from primary medical care centers or referral hospitals, medical claims data from animal insurance companies, and datasets constructed specifically for research purposes.
- Line 84 - Thus, veterinary medical big data may very well represent an underutilized resource in human medical research.
- Lines 320 to 323 - These large-scale datasets constructed with the express intent to understand the biological and environmental determinants of health and disease in companion dog populations in order to apply those findings to other species will become increasingly valuable to human medical research as the datasets grow in terms of content and population.
- Lines 326 to 327 - Use of medical data collected in dog populations will continue to be a rich source of information that can be used to inform our understanding of human health, longevity, and disease outcomes.
Author Response
Thank you so much for taking the time to review this manuscript again. Please see our specific comments below.
- Lines 21 to 22 - Large scale datasets such as this can be used to inform our understanding of human health, disease, and how to increase healthy lifespan.
This has been changed to "Large-scale datasets such as this can be used to inform our understanding of health, disease, and how to increase healthy lifespan for both dogs and humans."
- Lines 25 to 26 - Thus, data related to dog health presents many opportunities to discover insights into human health and disease outcomes. Here we describe several sources of veterinary medical big data that can be used human in research.
This has been changed to: "Thus, data related to dog health presents many opportunities to discover insights into human health and disease outcomes. Here we describe several sources of veterinary medical big data that can be used in translational research"
- Lines 27 to 28 - These sources include veterinary medical records from primary medical care centers or referral hospitals, medical claims data from animal insurance companies, and datasets constructed specifically for research purposes.
This change has been made as suggested.
- Line 84 - Thus, veterinary medical big data may very well represent an underutilized resource in human medical research.
This has been changed to: "Thus, veterinary medical big data may very well represent an underutilized resource in translational medical research."
- Lines 320 to 323 - These large-scale datasets constructed with the express intent to understand the biological and environmental determinants of health and disease in companion dog populations in order to apply those findings to other species will become increasingly valuable to human medical research as the datasets grow in terms of content and population.
This has been changed to: "...valuable to translational medical research as the datasets grow in terms of content and population."
- Lines 326 to 327 - Use of medical data collected in dog populations will continue to be a rich source of information that can be used to inform our understanding of human health, longevity, and disease outcomes.
This has been changed to: "Use of medical data collected in dog populations will continue to be a rich source of information that can be used to inform our understanding of health, longevity, and disease outcomes in dogs and humans."